

# Exploratory and confirmatory factor analyses identify three structural dimensions for measuring physical function in community-dwelling older adults

Guiping Jiang[1,2], Xiaohuan Tan[2], Hailong Wang[3], Min Xu[3] and Xueping Wu[2]

[1] School of Physical Education, Harbin University, Harbin, Heilongjiang, China
[2] School of Physical Education, Shanghai University of Sport, Shanghai, China
[3] Shangti Health Technology (Shanghai) Co., Ltd., Shanghai, China

Corresponding author
Xueping Wu, wuxueping@sus.edu.cn

## ABSTRACT

**Background**. Physical function is a strong indicator of biological age and quality of life among older adults. However, the results from studies exploring the structural dimensions of physical function are inconsistent, and the measures assessed vary greatly, leading to a lack of comparability among them. This study aimed to construct a model to identify structural dimensions that are suitable and best assess physical function among community-dwelling adults 60–74 years of age in China.

**Method**. This study was conducted in 11 communities in Shanghai, China, from May to July 2021. A total of 381 adults 60–74 years of age were included in the study. Measured physical function data were used in factor analyses. Data collected from individuals were randomly assigned to either exploratory factor analysis (EFA) ($n = 190$) or confirmatory factor analysis (CFA) ($n = 191$). The statistical software used in the study was SPSS for EFA and AMOS for CFA. To test the properties of the structural dimension model of physical function, various fit indices, convergent validity, and discriminant validity were calculated.

**Results**. The EFA results derived seven indicators in three factors, with 58.548% of the total variance explained. The three factors were mobility function (three indicators), which explained 26.380% of the variance, handgrip strength and pulmonary function (two indicators), which explained 19.117% of the variance, and muscle strength (two indicators) which explained 13.050% of the variance. The CFA indicated that this model had an acceptable fit ($\chi^2$/df ratio, 2.102; GFI, 0.967; IFI, 0.960; CFI, 0.959; and RMSEA, 0.076), and the criteria for convergent validity and discriminability were also met by the model.

**Conclusion**. The constructed structural dimension model of physical function appeared to be a suitable and reliable tool to measure physical function in community-dwelling adults aged 60–74 years in China. The structural dimension indicators identified by this model may help sports medicine experts and healthcare providers offer more targeted interventions for older adults to reverse or slow the decline of physical function and to offer actionable targets for healthy aging in this population.

# INTRODUCTION

Global aging is one of the defining challenges of this century (*National Academy of Medicine, 2022*). The World Health Organization estimates that the number of adults older than 60 years will nearly double globally by 2050, resulting in older adults outnumbering younger people worldwide for the first time (*World Health Organization, 2015*). The population in China is also rapidly aging. According to the seventh census of China, the number of people 60 years of age or older in 2020 was approximately 264 million, accounting for 18.7% of the total population in China. Thus, healthy longevity—extending the length of time an individual can live without limitations in daily activities—is becoming a top public health priority (*The US Department of Health and Human Services: Healthy People, 2014*; *World Health Organization, 2015*). In the 2015 *World Report on Ageing and Health*, the World Health Organization set a goal of healthy aging to help people develop and maintain their functional capacity (*World Health Organization, 2015*).

Physical function, as an intrinsic ability, is an objective measure of the ability to perform simple and complex activities of daily living (*Freiberger et al., 2012*; *Beaudart et al., 2019*) and is a strong indicator of biological age and a biomarker of health and quality of life among older adults (*Pavasini et al., 2016*; *Vestergaard et al., 2009*; *Cesari et al., 2009*; *Halaweh et al., 2015*). Both cross-sectional and longitudinal studies have shown that decreased physical function is associated with adverse health outcomes, such as decreased cognitive function (*Blankevoort et al., 2013*; *Taekema et al., 2012*), mobility limitations, disability in activities of daily living (*Donoghue et al., 2014*; *Cesari et al., 2009*; *Taekema et al., 2010*), hospitalization (*Donoghue et al., 2014*; *Campbell, Seymour & Primrose, 2004*), and increased mortality (*Justice et al., 2016*; *Pavasini et al., 2016*). Studies have found that physical function begins to decline in about 20% of healthy adults aged 60 years or older, and by age 80, physical function decreases are observed in about half of these adults (*Ervin, 2006*).

Physical function encompasses multiple dimensions, including among many other components, muscle strength, balance, endurance, and coordination (*Beaudart et al., 2019*; *Liu et al., 2014*). Given the many dimensions, it is perhaps unsurprising that there is a lack of uniformity for the structural dimensions that have been investigated across studies. In addition, the types of measures used to assess each dimension have varied greatly across studies, leading to a lack of comparability among studies assessing the same issue. Classic assessment of physical function in older adults has been conducted using the Short Physical Performance Battery, which consists of a 4 m walk at the usual pace, five sit-to-stand tests, and three standing balance tests (bipedal, semi-tandem standing, and tandem standing). However, studies have shown that this battery is associated with a ceiling effect in populations of community-dwelling healthy adults 60 years of age or older (*Power et al., 2014*; *Bergland & Strand, 2019*). Another assessment used for older adults is

the Senior Fitness Test, which consists of the 30 s chair stand test (CST), 30 s arm curl test (ACT), 6 min walk test (which can be replaced by the 2 min step test), back scratch test (BST), chair sit and reach test (CSRT), the Timed Up and Go (TUG) test, and an assessment of body mass index. Some studies have used single indicators, such as walking speed (*Li et al., 2021*), handgrip strength (*Legdeur et al., 2020*; *Stephan et al., 2013*), and the TUG test (*Garber et al., 2010*) to represent physical function, whereas other studies have used two or a few indicators to represent physical function. The use of a single or only a few indicators may be related to the challenge in collecting more comprehensive indicators of physical function in large samples. However, it is difficult to capture complete information about physical function with only a few tests (*Dansereau et al., 2020*) because different underlying mechanisms contribute to the decline of different indicators of physical performance. Moreover, decline in each indicator of physical function is heterogeneous (*Stephan et al., 2013*). Therefore, multiple measures are needed to assess physical performance in older adults (*Hoekstra et al., 2020*). The baseline age of participants in most previous studies has been 75 years or older (*Stephan et al., 2013*); thus, there is a lack of research on the structural dimensions of physical function indicators in the population of adults aged 60 to 74 years. Hence, it is necessary to identify which structural dimensions are most appropriate for the assessment of the physical function of Chinese adults aged 60 to 74 years. Such knowledge would enable sports medicine and fitness experts to propose the most appropriate interventions for adults in this population to effectively prevent or reverse the deterioration of specific physical functions. Those more specific interventions may help older adults in this age group to achieve healthy longevity and to maintain an independent lifestyle for a longer time (*Freiberger et al., 2012*).

Thus, this study aimed to construct a model to identify the structural dimensions of physical function that are suitable and that best assess physical function in community-dwelling adults aged 60 to 74 years in China. We hypothesized that mobility and muscle and pulmonary functions may be key structural dimensions of physical function in this population.

## MATERIAL AND METHODS

### Participants

This study recruited adults from 11 communities in Shanghai using a convenience sampling method. It was conducted from May to July 2021. This study was approved by the Ethics Committee of Shanghai University of Sport (102772021RT067). All participants provided written informed consent. The inclusion criteria were 60–74 years of age; ability to communicate normally and complete the study test independently; and voluntary participation in this assessment. The exclusion criteria were severe visual, hearing, or speech impairments; disease that may seriously affect test results (e.g., stroke, Parkinson's syndrome, osteoarthritis, severe pain, congestive heart failure, dizziness, severe respiratory disease, and mental disorders); and uncontrolled hypertension (systolic/diastolic blood pressure >160/100 mmHg). The participants included 381 adults aged 60–74 years. Of them, the group used for exploratory factor analysis (EFA) comprised 190 adults (61 men

and 129 women) with a mean age of 66.99 years, and the group used for confirmatory factor analysis (CFA) comprised 191 adults (57 men and 134 women) with a mean age of 67.06 years.

## Measurements

Test measurements included height, weight, and waist and hip circumference, with derived measures of body mass index (BMI) and waist-to-hip ratio (WHR), as indicators of body composition. Spirometry and the 2-min step test were conducted to indicate cardiorespiratory fitness. Handgrip strength, ACT and CST were used as indicators of muscle function. BST and CSRT were considered flexibility indicators. The one-legged stance (OLS) with eyes closed and the TUG test were used as indicators of balance. The 6 m walk test was used as an indicator of walking speed. The methods used to determine height, weight, waist circumference, hip circumference, spirometry, handgrip strength, and OLS were conducted as described previously (*Zhang, He & Xu, 2017*), as were the methods used to conduct the step test, ACT, CST, TUG, BST, and CSRT (*Rikli & Jones, 2013*).

The 6 m walk is a common method for assessing walking speed (*Aoyagi et al., 2001*). Participants walked more than 10 m from the starting position to the end point at both their usual walking speed and their maximal walking speed (as fast as possible without running). The starting position and the 2 m, 8 m, and 10 m positions were marked. The walking time was recorded with a stopwatch (accurate to the nearest 0.01 s) for 6 m of walking between 2 m to 8 m to avoid the effect of acceleration at the beginning of the first 2 m and deceleration by braking in the last 2 m on speed. Walking speed was calculated as 6 m divided by the time and was accurate to 0.01 m/s (*Jiang & Wu, 2022*).

## Statistical analysis

In this study, an EFA was conducted using IBM SPSS 26.0 to assess the structural dimensions of physical function. The maximum likelihood estimation was selected for factor extraction because it allowed for evaluations of model fit and cross-validation with the CFA (*Goretzko, Pham & Bühner, 2021*). Varimax rotation was utilized for factor rotation because it allows for easier interpretation of the results and replication in future samples (*Leech, Barrett & Morgan, 2015*).

The CFA was conducted using AMOS 24.0 to further examine the structural dimensions of physical function. Model fit was examined using the $\chi 2/df$ ratio, root mean square error of approximation (RMSEA), goodness of fit index (GFI), comparative fit index (CFI), and Tucker–Lewis Index (TLI). We tested the convergent validity of the structural dimension of physical function by assessing the average variance extracted (AVE), and we tested the discriminant validity by using the positive square root of the AVE and the correlation coefficient between the factors.

## RESULTS

### Descriptive statistics of participants' physical function indicators

Descriptive statistics for participant physical function indicators (in the form of means ± standard deviations) are given in Table 1. Data for each physical function indicator used in the EFA and CFA were not statistically different.

**Table 1  Participant demographic characteristics and measures of indicators of physical function.**

| Measure | Total ($n = 381$) | | EFA group ($n = 190$) | | CFA group ($n = 191$) | | |
|---|---|---|---|---|---|---|---|
| | Mean | SD | Mean | SD | Mean | SD | p |
| Height (m) | 1.62 | 0.07 | 1.62 | 0.08 | 1.62 | 0.06 | 0.799 |
| Weight (kg) | 62.12 | 8.90 | 62.34 | 8.95 | 61.90 | 8.87 | 0.633 |
| BMI (kg/m$^2$) | 23.61 | 2.83 | 23.67 | 2.84 | 23.54 | 2.82 | 0.650 |
| WC (cm) | 86.05 | 8.38 | 86.58 | 8.74 | 85.53 | 8.00 | 0.224 |
| HC (cm) | 97.16 | 6.05 | 97.50 | 6.29 | 96.63 | 5.80 | 0.279 |
| WHR | 0.89 | 0.06 | 0.89 | 0.06 | 0.88 | 0.06 | 0.472 |
| Spirometry (mL) | 1927 | 740 | 1952 | 784 | 1902 | 694 | 0.505 |
| step test (No.) | 104.78 | 13.79 | 104 | 15 | 106 | 13 | 0.155 |
| HS (kg) | 28.2 | 7.0 | 28.1 | 7.0 | 28.3 | 7.0 | 0.732 |
| ACT (No.) | 20 | 4 | 20 | 4 | 20 | 4 | 0.956 |
| CST (No.) | 18 | 4 | 18 | 4 | 18 | 5 | 0.330 |
| BST (cm) | −3.46 | 9.82 | −4.35 | 9.89 | −2.58 | 9.70 | 0.079 |
| CSRT (cm) | 1.32 | 10.16 | 1.11 | 10.26 | 1.52 | 10.08 | 0.692 |
| OLS (s) | 32.22 | 21.79 | 31.19 | 22.02 | 33.25 | 21.56 | 0.357 |
| TUG (s) | 0.91 | 0.13 | 0.91 | 0.14 | 0.92 | 0.13 | 0.539 |
| UWS (s) | 1.43 | 0.27 | 1.42 | 0.26 | 1.44 | 0.27 | 0.408 |
| MWS (s) | 1.84 | 0.31 | 1.81 | 0.32 | 1.87 | 0.30 | 0.079 |

Notes.

SD, standard deviation; BMI, body mass index; WC, waist circumference; HC, hip circumference; WHR, waist to hip ratio; HS, handgrip strength; CST, 30-s chair stand test; ACT, 30-s arm curl test; BST, back scratch test; CSRT, chair sit and reach test; OLS, one-legged stance; TUG, Timed Up and Go; UWS, usual walking speed; MWS, maximum walking speed; EFA, exploratory factor analysis; CFA, confirmatory factor analyses.

## Exploratory factor analysis

Thirteen indicators of physical function (BMI, WHR, spirometry, step test, handgrip strength, ACT, CST, BST, CSRT, OLS, TUG, usual walking speed (UWS), and maximum walking speed (MWS) were subjected to the EFA and Bartlett's test of sphericity. The Kaiser–Meyer–Olkin (KMO) test was used to demonstrate the adequacy of the sample size and suitability for factor analysis. According to the criteria of *Hair et al. (2010)* for factor structure, the eigenvalues should be >1.0, and the factor loadings should be >0.5. Therefore, to make the physical function structural dimension a well-defined factor structure, we retained indicators with factor loadings >0.50 and no double loading between any two indicators (*Liao, Huang & Wang, 2022*).

After conducting the EFA, seven indicators (spirometry, handgrip strength, ACT, CST, TUG, UWS, and MWS) and three factors were identified: mobility function, handgrip strength and pulmonary function (HSPF), and muscle strength. The KMO value of 0.616 exceeded the reference value of 0.6 and is generally considered to be very good (*Kaiser, 1974*). Bartlett's test of sphericity was significant ($\chi^2 = 363.367$, $p = 0.000 < 0.05$) (*Bartlett, 1951*), indicating that those data satisfied the conditions for the EFA. These three factors explained 58.548% of the variance (Table 2). Factor 1, with three indicators of mobility function (TUG, UWS and MWS), explained 26.380% of the variance. Factor 2, with two indicators of HSPF (handgrip strength and spirometry), explained 19.117% of the variance.

**Table 2** Factor loadings matrix after rotation.

| Variable | Factor 1: Mobility function | Factor 2: HSPF | Factor 3: Muscle strength |
|---|---|---|---|
| MWS | **0.907** | 0.055 | 0.047 |
| UWS | **0.903** | 0.007 | 0.015 |
| TUG | **0.668** | 0.053 | 0.256 |
| HS | 0.078 | **0.879** | 0.098 |
| Spirometry | 0.006 | **0.874** | −0.034 |
| CST | −0.018 | −0.084 | **0.882** |
| ACT | 0.303 | 0.183 | **0.752** |
| Eigenvalues | 2.466 | 1.510 | 1.212 |
| % of variance | 26.380 | 19.117 | 13.050 |

**Notes.**

Total variance explained is 58.548%.

MWS, maximum walking speed; UWS, usual walking speed; TUG, Timed Up and Go; HS, handgrip strength; CST, 30-s chair stand test; ACT, 30-s arm curl test; HSPF, handgrip strength and pulmonary function.

Bold font indicates the factor with the highest loading.

Factor 3, with two indicators of muscle strength (ACT and CST), explained 13.050% of the variance.

## Confirmatory factor analysis

The CFA was performed using a sample independent of the EFA sample to assess the factor structure of the physical functions identified in the EFA described above. We conducted CFA using AMOS 24.0 and assessed model fit indices ($\chi 2$/df ratio, GFI, IFI, CFI, and RMSEA) to examine how well the EFA-derived structural dimensions of physical function fit the data (see Table 3). A $\chi 2$/df ratio below 5 is considered a good model fit (*Hooper, Coughlan & Mullen, 2008*). An RMSEA index less than 0.08 is considered acceptable, and a GFI above 0.90 is considered good (*Hu & Bentler, 1999*). An IFI and a CFI higher than 0.95 are considered excellent (*Bentler, 1990*; *Schermelleh-Engel & Moosbrugger, 2003*). The factor model in this study had a good fit ($n = 190$; $\chi 2/ df = 2.102$, $p = 0.017 < 0.05$; GFI $= 0.967$, IFI $= 0.960$, CFI $= 0.959$; RMSEA $= 0.076$). Figure 1 shows the standardized factor loadings for the factor models. All factor loadings were statistically significant ($p$ values $< 0.001$).

Convergent validity refers to the degree of similarity of measurement results when different measures are used to assess the same characteristic. Convergent validity can be judged by assessing the average variance extracted and construct reliability (CR). Convergent validity is verified when the AVE value of a factor is greater than 0.50 (*Liao, Huang & Wang, 2022*). A CR $> 0.7$ indicates that the model has good convergent validity. In our analyses, the AVE and CR of the model's convergent validity (Table 3) met those requirements (although the CR for the factor muscle strength was slightly below 0.7 at 0.682). These results indicated that the measures highlight the qualities of the dimensional constructs and have internal consistency and that the model had good convergent validity.

The discriminant validity aspect of construct validity is satisfied when different methods are applied to measure different constructs and the observed values are distinguishable from one another. The square root value of the AVE was calculated and compared with

**Table 3  Model convergent validity and construct reliability.**

| Paths | | | Estimate | AVE | CR |
|---|---|---|---|---|---|
| UWS | <— | Mobility function | 0.901 | | |
| MWS | <— | Mobility function | 0.741 | 0.503 | 0.733 |
| TUG | <— | Mobility function | 0.383 | | |
| HS | <— | HSPF | 0.995 | 0.639 | 0.765 |
| Spirometry | <— | HSPF | 0.537 | | |
| CST | <— | Muscle strength | 0.626 | 0.521 | 0.682 |
| AST | <— | Muscle strength | 0.806 | | |

**Notes.**
UWS, usual walking speed; MWS, maximum walking speed; TUG, Timed Up and Go; HS, handgrip strength; CST, 30-s chair stand test; ACT, 30-s arm curl test; HSPF, handgrip strength and pulmonary function. Arrows indicate correction index of regression weights between variables; AVE, average variance extracted; CR, construct reliability.

the corresponding Pearson's correlation coefficient between the factors. Discriminant validity is obtained when the positive square root of the AVE of the factor is higher than its correlation with other factors (*Liao, Huang & Wang, 2022*). The correlation coefficients among the three dimensions of physical function for the older adults in this study (mobility function, muscle strength, and HSPF) ranged from 0.11 to 0.45 (Fig. 1, Table 4), suggesting a low to moderate positive correlation and indicating that the three factors varied in the same direction but were independent of one another, meeting the requirements of model discriminant validity.

## DISCUSSION

The aim of this study was to construct a model to identify structural dimensions that best assess physical function in community-dwelling adults 60–74 years of age in China to ensure early and effective interventions of physical function decline in this population.

A priori hypothesis models with three factors (mobility function, HSPF, and muscle strength) were constructed using EFA. For this assessment, Bartlett's spherical test values should be statistically significant, with a *p*-value less than 0.05 (*Bartlett, 1951*). The Bartlett's test for sphericity value in this study was less than 0.01. In addition, KMO values above 0.60 are considered acceptable, a value between 0.70 and 0.80 is fair, a value between 0.80 and 0.90 is good, and a value above 0.90 is considered perfect (*Kaiser, 1974*). For the physical function structural dimensions assessed in the present study, the KMO value of 0.616 was considered acceptable. Thus, the structural model assessed in this study had the required sample size and was suitable for factor analysis. The failure of the KMO value in this study to reach above 0.7 may be related to the sample size, which was limited due to the impact of COVID-19. Future studies should increase the sample size.

We used CFA to assess whether the data set fit the structural dimension model constructed by the EFA (*Menezes et al., 2019*). After applying EFA, 13 physical function indicators did not satisfy the requirement of factor loadings being greater than 0.5. Through continuous correction, the final seven indicators were divided into three factors that explained 58.548% of the variance: mobility function (three indicators), HSPF (two indicators), and muscle strength (two indicators). The factor loadings in the EFA results

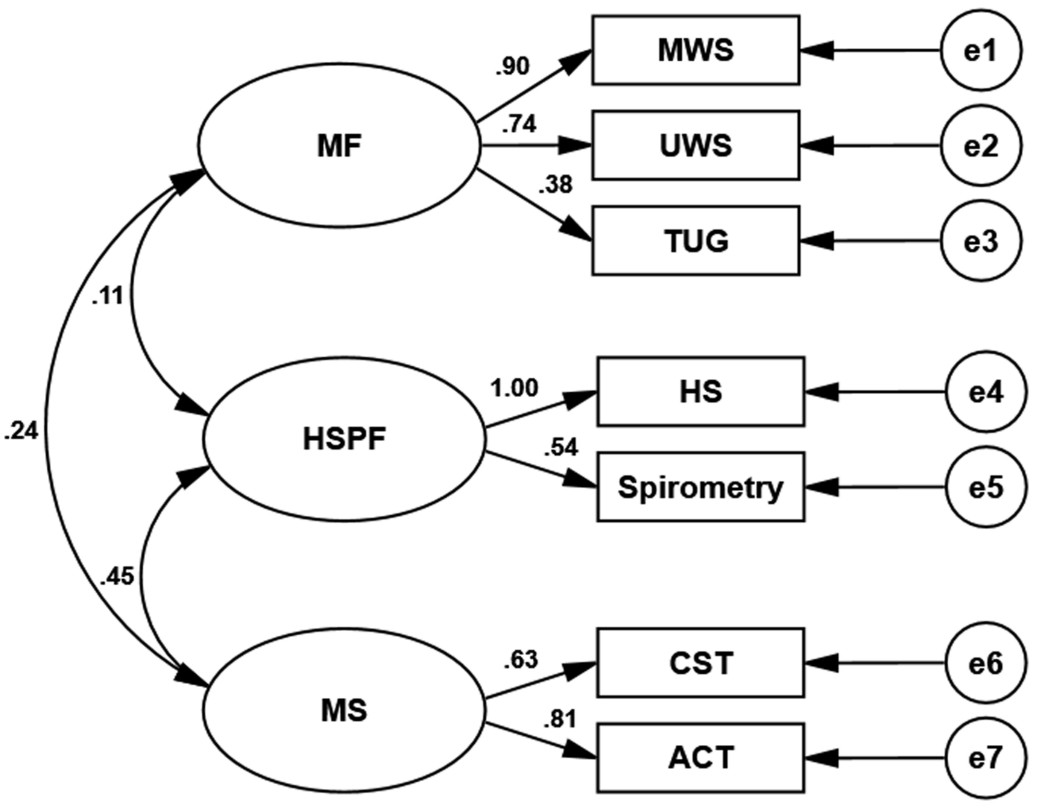

**Figure 1** **Schematic diagram of the relationship between physical function measures and metric factors in older adults.** Values next to arrows indicate standardized path coefficients. MF represents mobility function; HSPF, handgrip strength and pulmonary function; MS, muscle strength; MWS, maximum walking speed; UWS, usual walking speed; TUG, Timed Up and Go; HS, handgrip strength; CST, 30-s chair stand test; ACT, 30-s arm curl test, and e1–e7, exogenous variables 1–7.

**Table 4** **Discriminant validity of the structural dimension of physical function.**

| Component | Mobility function | HSPF | Muscle strength |
| --- | --- | --- | --- |
| Mobility function | 0.503 | – | – |
| HSPF | 0.115 | 0.639 | – |
| Muscle strength | 0.240 | 0.448 | 0.521 |
| $\sqrt{\mathrm{AVE}}$ | 0.709 | 0.800 | 0.722 |

**Notes.**
  HSPF, handgrip strength and pulmonary function; $\sqrt{\mathrm{AVE}}$, square root values of the average variance extracted.

ranged from 0.668 to 0.907, all greater than the value of 0.5 suggested by *Hair et al.( 2010)*. These results indicated that the structural dimension had a clear factor structure. The first factor, mobility function, explained 26.380% of the variance. Mobility function is based on muscle strength, balance, endurance, and coordination (*Wang & Chen, 2021*). Mobility depends not only on the ability to walk but also on the ability to maintain stability while standing and to control the transition from one posture to another (*Tyson & Connell, 2009*). Walking relies heavily on dynamic stability control, and as walking speed increases,

so do the requirements for strength and range of motion (*Winter, 1995*). Thus, although there is overlap in the musculoskeletal and central nervous system requirements for each index of mobility function (e.g., UWS, MWS, and TUG), the combination of requirements is unique (*Cesari et al., 2009*). Therefore, in this study, mobility function was measured using three different tasks—UWS, MWS, and TUG—to potentially identify the different contributions of the different systems. The second factor, HSPF, explained 19.117% of the variance. Pulmonary function has been found to be linearly correlated with physical performance based on handgrip strength and CST tests in healthy community-dwelling populations (*Landi et al., 2020*). Handgrip strength has also been independently correlated with spirometry even for healthy Han Chinese older adults (*Chen et al., 2020*), which is consistent with the results of the present study. The third factor, muscle strength, explained 13.050% of the variance. Considerable evidence suggests that the ability to perform physical tasks is determined by threshold levels of muscle strength and endurance (*Brown, Sinacore & Host, 1995*; *Buchner et al., 1992*; *Buchner, 1997*; *Evans, 1995*). Individuals who lack the necessary strength may not be able to perform the various activities of daily living that are important determinants of independence (*Brill et al., 2000*). Muscle strength is important for maintaining physical function in older people (*Chiung-ju et al., 2014*), and leg strength is highly correlated with physical performance tests (*Reid et al., 2014*), findings consistent with those of the present study. We also found a positive correlation between upper limb muscle function and physical function performance. Therefore, assessing upper limb muscle function, which is required for pushing, pulling, catching, and lifting in daily life, will provide a more comprehensive understanding of the overall physical function of older people. A systematic review has shown that baseline muscle measurements are predictors of the future ability to perform activities of daily living and instrumental activities of daily living dependence in older adults (*Wang et al., 2020*). Decreased muscle strength and power are associated with reduced functional capacity and mobility and poor health outcomes (*Brown, Sinacore & Host, 1995*; *Dela & Kjaer, 2006*). These findings are consistent with the results of present study showing that muscle strength is significantly and positively correlated with mobility function and HSPF as an important dimension of physical function.

The model fit index of CFA met the requirements (*Hooper, Coughlan & Mullen, 2008*; *Bentler, 1990*; *Hu & Bentler, 1999*) and indicate an acceptable model fit. Regarding convergent validity, the AVE values ranged between 0.503 and 0.639, meeting the suggested cutoff value of 0.50 (*Liao, Huang & Wang, 2022*). The CR values ranged between 0.682 (approximately equal to 0.7) and 0.765, meeting a suggested cutoff value of 0.70 (*Hair et al., 2010*). Discriminant validity met the criteria of *Hair et al. (2010)*, *Liao, Huang & Wang (2022)* and *Fornell & Larcker (1981)*. Three factors—mobility function, HSPF, and muscle strength—were identified as different dimensions of physical function in this population of adults 60–74 years of age. Mobility function appears to be the most important dimension of physical function in older adults, followed by HSPF and muscle strength. These factors were all independent but interrelated, indicating that they were subsumed under bodily function but were focused on different aspects of bodily functions. The consideration of multiple influences in more complex domains when assessing and intervening in physical

function may facilitate early multidimensional identification of physical function decline in older adults and may provide targeted intervention strategies.

This study had some limitations. First, the sample population in this study comprised community-dwelling adults aged 60–74 years residing in the Shanghai area of China, limiting the application of the results to other geographical regions or other age groups. The sample size of this study was limited due to the COVID-19 pandemic. Second, the physical function indicators used in this study were based on a combination of ease of use, portability of equipment, time, costs, and reliability and thus all potential indicators were not included. In addition, only two variables were included in both the muscle strength and HSPF metrics, which may have affected the results. Third, this study was conducted to investigate potential relationships among latent variables through the cross-sectional assessment of physical functioning data. Future studies are required to determine causal relationships among latent variables through longitudinal studies or experimental studies.

## CONCLUSION

Given that physical function is a biomarker of health and quality of life in older adults, this study aimed to construct a model to identify structural dimensions that best assess physical function in community-dwelling older adults in China. The results indicated that the structural dimension model constructed in this study appeared to be a reliable tool to measure physical function in this population. These structural dimension indicators may help sports medicine experts and healthcare providers to offer more targeted interventions for older adults to reverse or slow declines in physical function and to offer actionable targets for healthy aging.

## ACKNOWLEDGEMENTS

We are grateful to Shangti Health Technology (Shanghai) Co., Ltd. for their support in recruiting participants, and to Prof. Wu's research group for assistance in data collection.

### Funding

This study was supported by The Program for Overseas High-Level Talents at Shanghai Institutions of Higher Learning (No. TP2020063) and the Heilongjiang Province Key Commissioning Project (SJGZ20200098). The funders had no role in study design, data collection and analysis, decision to publish, or preparation of the manuscript.

### Grant Disclosures

The following grant information was disclosed by the authors:
The Program for Overseas High-Level Talents at Shanghai Institutions of Higher Learning: TP2020063.
The Heilongjiang Province Key Commissioning Project: SJGZ20200098.

## Competing Interests

Hailong Wang and Min Xu are employed by Shangti Health Technology (Shanghai) Co., Ltd.

## Author Contributions

- Guiping Jiang conceived and designed the experiments, analyzed the data, prepared figures and/or tables, authored or reviewed drafts of the article, and approved the final draft.
- Xiaohuan Tan and Min Xu performed the experiments, analyzed the data, prepared figures and/or tables, and approved the final draft.
- Hailong Wang and Xueping Wu conceived and designed the experiments, authored or reviewed drafts of the article, and approved the final draft.

## Human Ethics

The following information was supplied relating to ethical approvals (i.e., approving body and any reference numbers):

This study was approved by the Ethics Committee of Shanghai University of Sport (102772021RT067).

## Data Deposition

The raw data are available in the Supplementary Files.

## Supplemental Information

Supplemental information for this article can be found online at http://dx.doi.org/10.7717/peerj.15182#supplemental-information.

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
