# Peer review of "Exploratory and confirmatory factor analyses identify three structural dimensions for measuring physical function in community-dwelling older adults"

_PeerJ, doi:10.7717/peerj.15182_

## Round 0.1 · original submission · Major Revisions

Dear Authors,

Please reply point by point to the reviewers' comments.

Reviewer 1 ·

Basic reporting

The article is well-written and clear. I am not sure if it was a restraint of the formatting requirements, but it would be helpful if the table column and row titles could be on single lines rather than wrapped to a second line. This would help maintain alignment of the data and assist the reader in understanding the table more easily/quickly.

Experimental design

I have a few questions for the authors regarding the analysis.

1. The authors indicate they used principal components analysis for their EFA. Are you referring to principal components factor analysis or principal components analysis? Why was this analysis technique chosen compared to other options (e.g., maximum likelihood extraction)? What type of rotation was used? I do not know that the authors’ explanation needs to be incorporated into the manuscript, but it would be helpful to understand their reasoning behind their current analysis choice. Clarity as to whether PCA or principal components factor analysis was used would be helpful to include in the manuscript.

2. What cut-off values were used to assess the EFA? For example, what was the minimum acceptable loading for an item? What was the highest acceptable cross-loading? In some cases, cross-loadings of ≥0.30 indicate that the item should be removed (Leech, Barrett, & Morgan, 2015). Under these guidelines, item ACT may not be appropriate to include in the final solution.

3. Can you please clarify why you used 0.90 as the cut-off values for your global fit indices when assessing the CFA. The most recent recommendations often suggest using a cut-off value of ≥0.95 for many global fit indices (e.g., CFI, etc.; Hu & Bentler, 1999). Furthermore, while ≤0.08 is acceptable for RMSEA, ≤0.06 has been considered more optimal (Hu & Bentler, 1999).

4. Were modification indices (MI) assessed during the CFA process? For example, was there a high modification index for either the handgrip strength or spirometry variable that could indicate needed modification of this factor?

Validity of the findings

No comments as the article is currently written. If any updates are made based on the previous comments, please update findings/conclusions accordingly.

Additional comments

Please double check in-text and reference list formatting. Sometimes there are missing spaces between author last name and year in-text (e.g., line 249), and inconsistencies in formatting in the reference list (e.g., year is bolded in some and not in others; space before article year, etc.)

Line 300: The word “older” needs to be capitalized as part of the title of the working group.

Line 347: The authors state that the two-item factor “could have affected the results”. While this is a valid point, please consider elaborating on how/why the results may have been affected.

Reviewer 2 ·

Basic reporting

The authors report on an investigation of physical function traits in elderly citizens in various populations in China.

In its recent form, the manuscript presents research such that aging is an issue in China, and it is not only a problem for China, but it will be a problem worldwide ("The population in China is rapidly aging and has become an urgent challenge not only for China but worldwide.." So, it seems that the problem is that China will have an impact worldwide. Surely, that is a misinterpretation due to suboptimal language use in the text. I recommend that the authors start presenting the global problem (i.e., aging as a world problem), then present China's case as an instance of this problem rather than reporting a problem in China.

The Introduction section does not present the motivation of the study effectively. "... there is an urgent need" does not have to be addressed by a scientific article. So, the authors are recommended to present the research question(s) of the study and explain the hypotheses based on previous work and research questions (if any). Why do we need those structural dimensions, why are they missing in the literature (maybe it is difficult to collect data rather than there exists an emergent gap that the authors noticed)?

The coverage of the previous work is somewhat outdated (the authors check recent work in the field).

Experimental design

Materials and Methods section has missing information, such as the number, gender, and age of the participants. It is surprising not to see this information in the Participants section but in another section. Demographic data is not part of the Results section.


The statistics report includes stylistic errors (such as the p-value in capital P). These are important clues that the authors should focus more on reporting statistics following specific guidelines.

It is not clear why the authors use EFA/CFA but not other methods. This does not mean that they should use other methods, but to emphasize that the motivation for using the reported statistics is not clearly presented.

Validity of the findings

The content presented in the Discussion section looks like a review of the literature rather than a discussion of the findings. It is expected that the authors discuss such factors that might have had an impact on the findings. For instance, given the date of data collection, COVID might have had an impact on the study. Overall, the study has merits. However, it is not efficiently presented in the manuscript.

Additional comments

Overall, it is difficult to read the article. There are too many ambugities in the sentences. There are also numerous misnomers, such as "actual data" (do they have non-actual data?). It is highly recommended that authors receive professional language editing support.

---

## Round 0.2 · accepted · Accept

The original Academic Editor is not available so I have taken over handling the submission.

The reviewers either declined to re-review or did not respond to my invitation but I confirm that the authors have addressed all of the reviewers' comments.